# A multivariate, quantitative assay that disentangles key kinetic parameters of primary human T cell function *in vitro*

**Grace L. Huang☉, Daniel P. Nampe☉, Jason Yi¤, Grant B. Gabrelow, Kathleen R. Negri, Alexander Kamb, Han Xu©***

Discovery Research, A2 Biotherapeutics, Inc., Agoura Hills, California, United States of America

☉ These authors contributed equally to this work.
¤ Current address: ImmPACT-Bio, Inc., Camarillo, California, United States of America
* hxu@a2biotherapeutics.com

**Data Availability Statement:** All relevant data are within the paper and its Supporting information files.

## Abstract

Cell therapy is poised to play a larger role in medicine, most notably for immuno-oncology. Despite the recent success of CAR-T therapeutics in the treatment of blood tumors and the rapid progress toward improved versions of both CAR- and TCR-Ts, important analytical aspects of preclinical development and manufacturing of engineered T cells remain immature. One limiting factor is the absence of robust multivariate assays to disentangle key parameters related to function of engineered effector cells, especially in the peptide-MHC (pMHC) target realm, the natural ligand for TCRs. Here we describe an imaging-based primary T cell assay that addresses several of these limitations. To our knowledge, this assay is the first quantitative, high-content assay that separates the key functional parameters of time- and antigen-dependent T cell proliferation from cytotoxicity. We show that the assay sheds light on relevant biology of CAR- and TCR-T cells, including response kinetics and the influence of effector:target ratio.

## Introduction

Cell therapy is an especially complex modality, involving two integrated components: a recombinant vector that provides the targeting function(s); and a cellular component that receives the vector and discharges effector functions [1, 2]. These two components interact to generate the full therapeutic product that, in the case of cancer treatment, clears the patient's body of tumor cells. For currently approved therapies such as Yescarta™ and Kymriah™, the cell component of the medicine is autologous, derived from each patient's peripheral blood mononuclear cells [3, 4]. These cells extracted from the body contain multiple cell types in different proportions from patients with disparate (i) ages; (ii) comorbidities; (iii) prior therapies; (iv) tumor burden; and, (v) HLA haplotypes. Typically, the patient's T cells are partially purified, transduced with a viral vector that encodes the targeting receptor, expanded, and reinfused into the patient. This manufacturing process introduces further sources of variation via gene transfer

**Funding:** This study was funded by A2 Biotherapeutics. A2 Biotherapeutics provided support in the form of salary for authors GLH, DPN, KRN, JY, GBG, AK, and HX. The funders had no role in study design, data collection and analysis, decision to publish, or preparation of the manuscript. The specific roles of these authors are articulated in the 'author contributions' section.

**Competing interests:** The authors have read the journal's policy and have the following competing interests: GLH, DPN, KRN, GBG, AK, and HX are paid employees of A2 Biotherapeutics. JY is a paid employee of ImmPACT-Bio. This does not alter our adherence to PLOS ONE policies on sharing data and materials. There are no patents, products in development or marketed products associated with this research to declare.

efficiency, expression levels, outgrowth of certain cell types, and so on. The net result engenders wide functional variability that is not well understood at this time [5, 6].

Further upstream in the discovery phase of the drug development process, selection of the signaling construct plays a critical role in ultimate success. Optimization of the receptor ideally involves a thorough understanding of not only the receptor's inherent features such as ligand-binding and signaling but also the impact of those properties on the function of effector cells. Much of this complex pharmacology and biology are characterized currently with relatively simple assays that comprise, for instance, mixing effector and target cells at specific ratios and measuring outputs such as cytotoxicity and cytokine production [7–10]. These assays cannot resolve different contributions to function, including rates of proliferation and killing.

We set out to devise an assay that can accommodate the increased complexity of primary T cell behavior, but has the relative simplicity and robustness of the workhorse Jurkat/T2 assay [11]. We developed a process based on quantitative imaging that tracks effector and target cells, and disentangles many important variables: Quantitative Imaging-based Killing (QuIK) assay. The QuIK assay measures several parameters over time and can deconvolute antigen-specific T cell kill rate from proliferation rate. It is well-suited to CAR- and TCR-Ts that recognize peptide-major histocompatibility-complex (pMHC) targets. The assay should prove useful to characterize fundamental aspects of CARs and TCRs that have heretofore not been routinely observable, as well as to analyze important attributes of scaled-up cell therapeutic products. This assay also illuminates relevant biology of CAR- and TCR-T cells which may affect selection and evaluation of therapeutics.

## Materials and methods

### Cell line culture method

Cells were cultured in a Heracell 150i incubator (Thermo Fisher Scientific) at 37°C and 5% $CO_2$. A375 (CRL-1619), A375 Firefly-luciferase (CRL-1619-LUC2), and Ca Ski (CRM-CRL-1550), were purchased from ATCC®. Renilla-luciferase was stably expressed in Ca Ski cells. These cells were cultured in Dulbecco's Modified Eagle Medium (Gibco 11965–092) supplemented with 10% fetal bovine serum. Jurkat NFAT-Firefly-Luciferase cells were purchased from BPS Bioscience (60621) and were cultured in RPMI1640 (Gibco 72400–047) supplemented with 10% heat-inactivated fetal bovine serum. T2 cells were also purchased from ATCC® (CTL-1992), and cultured in IMDM (Gibco 12440–053), supplemented with 20% heat-inactivated fetal bovine serum. 100 U/mL Penicillin-streptomycin (Gibco 15140163) (1X P/S) was used in all media. Adherent cells lines were passaged when they reached approximately 80% confluency. Suspension cells were kept below a density of 1E6/mL.

### Transduction and culturing of human primary T cells

Primary human PBMCs were purified from Leukopaks purchased from Allcells® according to the method described in Garcia, et al., 2014 [12]. Collection protocols and donor informed consent were approved by an Institutional Review Board (IRB) at Allcells®. Allcells® also followed HIPAA compliance and approved protocols with strict oversight (https://www.allcells.com/cell-tissue-procurement/donor-facilities/). Unless otherwise specified, all LymphoONE™ media (Takara WK552) was supplemented with 1% Human AB Serum (GeminiBio 100–512). Day 0: Approximately 500,000 thawed human PBMCs were grown in 0.5 mL LymphoONE™ and supplemented with TransAct™ (Miltenyi 130-111-160) following the manufacturers guidelines (1:100 dilution) and grown in a 24-well plate with 500 μL LymphoONE™. Day 1: Lentivirus encoding CAR or TCR was added at a MOI of 5. Day 2: 500 μL of LymphoONE™ supplemented with IL2 (300 IU/ml) was added (Fig 1A). Day 5: Cells were pelleted and

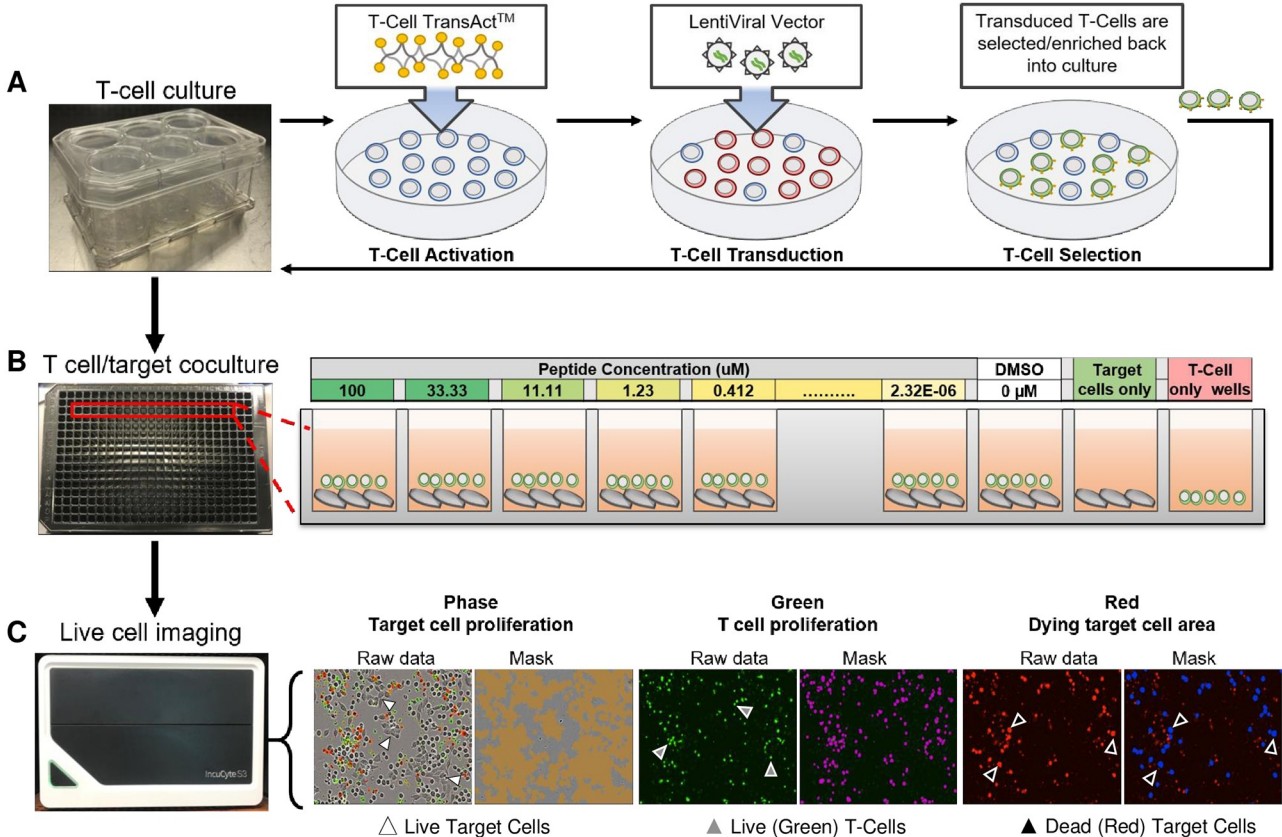

**Fig 1. Overview of QuIK assay.** (A) Schema of T cell transduction and culture as described in Materials and Methods. (B) Diagram of QuIK assay set up: tumor cells loaded with target peptide at different concentrations were cocultured in 384-well plates, and incubated in an IncuCyte® to monitor morphological changes. (C) Examples of readout: Target cell confluency was determined by phase contrast. Calculated confluent area is marked in orange. T cells are loaded with CellTracker™ Green, and live T cells (green-positive and red-negative) are marked in magenta. Dead/dying cells that incorporated Annexin V/PI are red. Dead/dying target cells were separated from T cells by size and shape and are marked in blue.

resuspend in 3 mL LymphoONE™ supplemented with IL2 (300 IU/mL) and transferred to a 24-well G-Rex plate (Wilson Wolf 80192M). Day 7: 1ml LymphoONE™ supplemented with IL2 (300 IU/mL) was added. Day 9: Cells were labeled with Protein L-biotin/streptavidin-PE or mTCR-PE. Labeled cells were then incubated with Anti-PE microbeads (Miltenyi 130-048-801) according to the manufacturer's instructions. Labeled cells were then enriched using AutoMACS® Pro Separator (Miltenyi). Positive cells were resuspended in 3 ml LymphoONE™ supplemented with IL2 (300 IU/mL) and grown in 24-well G-Rex. Fresh IL2 (300 IU/mL) was added every 48 hours with a media change every 7 days.

### Flow cytometry to determine T cell transduction efficiency and proliferation

The expression of CARs and TCRs in primary T cells were determined by flow cytometry using biotinylated protein L (ThermoFisher #29997) followed by fluorescently labeled strepta-vidin, or fluorescently labeled anti-murine TRBV antibody (Biolegend Cl:H57-597) (S1 Fig). All staining was carried out at 4˚C, and median fluorescence intensity (MFI) was determined using a FACS Canto II flow cytometer (BD Biosciences). For T cell proliferation, transduced T cells were stained with CMFDA (Invitrogen C7025; fresh CMFDA working solution at 1 µM

was made in LymphoONE™ from the solid) and mixed with 25% of untransduced T cells that were not loaded with CMFDA on day 0. Mixed T cells were cocultured with target cells while being imaged on IncuCyte® or ImageXpress® Micro (IXM) for 3 days. On day 3, T cells were recovered from the imaging plate. T cells before and after co-culture were stained with anti-CD3 antibody (Biolegend 300439) without washing. Relative T cell proliferation was calculated using the ratio of CD3+/CMFDA+ vs. CD3+/CMFDA- populations within each sample, and was compared among different peptide concentrations and E:T ratios (S5 Fig).

## Activation of Jurkat NFAT-luciferase cells cocultured with peptide-loaded T2 cells

On day 1, Jurkat NFAT-Firefly-Luciferase cells were transfected with the previously described TCR and CAR constructs using standard protocols for the Lonza 4D Nucleofector™ (AAF-1002B). T2 cells were loaded with the peptides indicated in Table 1. Peptide was resuspended in DMSO, and serially diluted 3-fold twenty times. Serially diluted peptide solutions were added to T2 cells resuspended in peptide-loading media (RPMI1640 + 1% BSA + 1X P/S). This yielded peptide-loaded T2 target cells at approximately 1E6/mL, with peptide concentrations ranging from ~10 fM to 100 μM, including a control at 0 μM. Peptide-loaded T2 cells were incubated overnight at 37˚C in 384-well plates (Thermo Scientific AB0781). On day 2, the cells were cocultured in a 384 well plate (Corning 3570). Peptide-loaded T2 cells (10,000 cells/well) were added to transfected Jurkat-NFAT-Firefly-Luciferase cells (12,000 cells/well) to a final volume of 20 μL. After a 6-hour incubation at 37˚C, 20 μL firefly luciferase reagent (BPS Bioscience 60690–1) was added to each well and incubated for 10 minutes in the dark. Luminescence was monitored on the Tecan Infinite® M1000 with a 100 ms integration time.

## Target cell killing by primary T cells, monitored by loss of luciferase signal

For peptide loading assay: on day 1, A375 firefly-luciferase cells were seeded into a 384 well plate at 2000 cells per well. These cells were peptide-loaded as described above in the Jurkat/T2 section. In this assay, the cells were peptide-loaded in LymphoONE™ supplemented with 1% human serum and 1X P/S. On day 2, a calibration curve was generated using CellTiter-Glo® (Promega G7570) readout to determine the number of cells seeded per well. From this curve, the appropriate number of T cells to add to each well was calculated. T cells were gently added to the peptide-loaded A375 firefly-luciferase cells at an E:T ratio of 3:1 in 20 μL. After 48 hours, 25 μL of firefly luciferase solution was added to each well, followed by a 10 minute integration in the dark. Bioluminescent signal was monitored on the Tecan Infinite® M1000 with a 100 ms integration time.

For cytotoxic T cell assay using Ca Ski (HPV+) or A375 (HPV-) as target cells: on day 1: Ca Ski-Renilla-Luciferase-GFP and A375-Firefly-Luciferase cells were plated into a 384-well plate. The target cells were plated at 2600 cells/well in LymphoONE™ supplemented with 1% human

**Table 1.  Receptor constructs used in this study.**

| Construct | Antigen peptide |
|---|---|
| HPV E6 TCR | TIHDIILECV |
| HPV E6 CAR | TIHDIILECV |
| HPV E7 TCR | YMLDLQPET |
| HPV E7 CAR | YMLDLQPET |

All are HLA-A*02-restricted.

AB serum and 1X P/S. Target cells were incubated overnight (18–24 hours) at 37˚C. On day 2: a calibration curve was generated to determine the accurate number of viable adherent target cells plated per well as described above. Primary T cells expressing the CAR or TCR (effector cells) were mixed with target cells at the following E:T ratios: (27:1, 9:1, 3:1, 1:1, 0.3:1, 0:1).

Primary T cell killing of target cells was quantified at 48 hours by adding 25 μL per well of either Renilla-Glo Luciferase (Promega E2720) or Firefly Luciferase, followed by 5 minute incubation before the luminescent signal was measured on the Tecan Infinite® M1000 with a 100 ms integration time.

## IFNɣ secretion by primary T cells

IFNɣ was measured by AlphaLISA according to the IFN-γ detection kits from PerkinElmer (AL217F). After 48 hours of co-culture, 3 μL of supernatant was collected and transferred into an AlphaLISA plate (ProxiPlate-384 Plus 6008280) to measure the IFNɣ concentration of all experimental conditions. IFNɣ standards were provided in the detection kit and the analyte was titrated to produce a wide dynamic range of signal that corresponds to the various concentration of IFNy (in pg/mL) present in the wells. The standards and samples were run in-parallel so that the signal of the samples could be interpolated to the standard curve to determine the IFNy concentration of each condition. The AlphaLISA assay was measured on the EnSpire® using standard Alpha settings.

GraphPad Prism was used to plot the standard curves using a nonlinear regression, four-parameter logistic equation to fit a sigmoidal curve as a function of dose-response. IFNy concentration was then determined by interpolating the AlphaLISA signal from each sample onto the standard curves; data shown are the average of duplicate samples.

## QuIK assay

Adherent target cells (A375) were cultured in standard conditions, and seeded at 2000 cells per well in LymphoONE™ media containing 1% human serum in 384 well plates (Greiner 781091). Serially diluted peptide solutions were added to the target cells to yield peptide concentration ranging from ~2 pM to 100 μM (Fig 1B). Then, 2000 peptide-loaded target cells were incubated at 37˚C overnight.

Fresh CMFDA working solution at 1 μM was made in LymphoONE™ from the solid. Percentage of TCR- and CAR-positive T cells, determined by flow cytometry, were used to adjust T cells numbers for co-culture with target cells. T cells were harvested and resuspended in CMFDA working solution at 200,000 cells/mL, and incubated in the dark at 37˚C for 30 minutes. After staining, cells were spun down and washed twice with 1 mL LymphoONE™ media containing 1% human serum. Cells were counted and added at an E:T ratio of 3:1, 1:1, or 1:3 to the peptide-loaded target cells. Additionally, Annexin V Red (Sartorius 4641) was resuspended according to manufacturer's instructions, and added to the coculture at a final dilution of 1:200. Propidium iodide was added at a final dilution of 1:10,000 (ThermoFisher P3566). Immediately after adding T cells to peptide-loaded target cells, the plate was placed in the IncuCyte® S3 and scanned every 2 hours or the ImageXpress® Micro (IXM) and scanned every 4 hours (S4 Fig) using phase, green, and red channels with 300 ms exposure for the fluorescent channels at 10x magnification.

## QuIK assay analysis algorithm

**Normalized T cell proliferation.** T cells were identified and counted using the IncuCyte® image analysis software based on size (5–15 μm), shape, and intensity of the CMFDA green dye that stained the cells. The software's built-in algorithm tracked and counted the

number of green T cells that were present in our field of view throughout the course of the experiment. From the raw counts, the T cells proliferative capacity (*P*, or fold increase) was quantified to demonstrate the T cells response over time with and without antigen stimulation:

$$P(t, c) = \frac{E_{live}(t, c)}{E_{live}(t_0 = 2, c)} \qquad (1)$$

where $E_{live}(t, c)$, is the T cell counts at a given time (t) and peptide concentration (c); $E_{live}(t_0 = 2\ hr, c)$ is the initial T cell count at the start of the co-culture. Since the co-culture was tracked every 2 hours for the duration of the experiment, we chose $t_0 = 2\ hr$ to allow time for the cells to settle at the bottom of the well for an accurate count. The data presented for proliferation is shown as fold increase to show the expansion of T cells over time relative to the initial cell number.

**Total target cell death and specific killing of target cells.** Total death of target cells was assessed by the percentage of red pixel area divided by the pixel area under phase-contrast:

$$T_{dead}(t, c) = \frac{\Sigma Red\ pixel}{\Sigma Phase\ pixel} \times 100\% \qquad (2)$$

$T_{dead}(t, c)$ is the fraction of dead targets cells at time t and peptide concentration c with respect to the target cell confluency per well. Therefore, the model captures the kinetics of both the growth and death of target cells as two dependent variables irrespective of the initial population variability from well to well. However, the increase in cell death includes both peptide-dependent and independent killing, or on-target and off-target CTL cytotoxicity. For this reason, specific killing of target cells (*K*) was estimated based on the total death of target cells at peptide concentration, c, normalized to the total death at 0 μM peptide. Eq (3) was used to determine specific killing:

$$K(t, c) = T_{dead}(t, c)/T_{dead}(t, 0) \qquad (3)$$

Specific killing (*K(t,c)*) is the ratio of on-target to off-target cell death that occurs for all tested CARs and TCR in this assay. The normalization to 0 μM peptide concentration serves as a control to distinguish the receptors' cross-reactivity to the target cells independent of the peptide. Therefore, if non-specific killing is evident, the value for specific killing at concentration c is expected to be similar to the 0 μM condition.

To estimate the killing efficacy on a per T cell basis, both the specific killing and T cell count were used as inputs to Eq (4):

$$S(t, c) = \frac{T_{dead}(t, c)/E_{live}(t - 6hr, c)}{T_{dead}(t, 0)/E_{live}(t - 6hr, 0)} \qquad (4)$$

where we defined *S(t, c)* as cytotoxic T cell activity that takes into account the T cell number from Eq (3). It is important to note that higher killing is evident when the data were examined at 6 hour intervals, so we estimate that the observed killing of target cells at time *t* is a result of the cumulative killing from the T cells that existed at (*t*– 6) hours. As such, a 6-hour time increment was used in all subsequent calculations.

**Proliferation rate (P').** Calculating the T cell proliferation rate is an estimated measure of cell division on a per T cell per day basis. The input for proliferation rate is the difference in T

cell counts at 6 hour intervals as shown in the equation below:

$$P\prime(t, c) = \left( \frac{E_{live}(t, c) - E_{live}(t - 6hr, c)}{E_{live}(t - 6, c)} \right) \times 4 \qquad (5)$$

Unlike Eq 1, which defines the overall cell expansion through time series data, Eq 5 estimates the extent of dividing cells occurring per day. We assume that T cell division could still occur in the absence of cytokines, but the presence of peptides would further increase their proliferation rate by on-target activation/stimulation. Therefore, tracking their proliferation rate demonstrates such activity in parallel with specific killing at different peptide concentrations.

**Specific kill rate (K').** Calculating the kill rate kinetics provides a model of T cell activity on a per hour basis from which we estimate the increase, decrease, and peak time of specific killing per T cell. Eq (6) was used to calculate the rate of target cell death per T cell per hour:

$$K\prime(t, c) = \left( \frac{K(t, c) - K(t - 6hr, c)}{E_{live}(t - 6hr, c)} \right) \div 6hr \qquad (6)$$

As Eq 4 depicts the overall specific killing throughout the culture time, Eq 6 describes the rate at which cytotoxic activity occurs and the time it takes to hit its maximum killing potential. Like the proliferation rate, we assume that the kill rate will differ with peptide concentration and T cell number. Therefore, plotting the kill rate helps reveal important parameters that are dependent on such conditions.

## Results

### Overall assay setup and choice of the key parameters to measure

We first identified the parameters most useful to classify and rank-order CAR- and TCR-T cells. To do this, we drew on our experience with Jurkat/T2 assays where we routinely measure variables in the context of a typical pMHC target which are relevant to sensitivity and specificity (EC50, Emax, etc.) [11]. Analogs of these parameters were defined and used for analysis. We focused on proliferation and cytotoxicity, as these are two interrelated *sine qua non* phenotypes for efficacy of infused primary human T cells [13].

The assay itself is based on the IncuCyte® instrument and uses three dyes and high-content microscopy to output the parameters described below. We spot-checked cytotoxicity and proliferation on the IXM (S4 Fig). In addition, we also confirmed T cell proliferation using flow cytometry (S5 Fig). The system was designed to distinguish live from dead effector and target cells, and report the percentage of dead/dying target cells and live target and effector cells (Fig 1). Normalization allowed calculation and visualization of specific killing by removing the contributions of allo-reactivity and/or nonspecific killing for each receptor construct. We were unable to estimate total target cell number directly, so we used pixel count under phase-contrast as a surrogate. The target cells could be distinguished from T cells based on size, and on the inclusion of a vital green dye (CellTracker™ Green, CMFDA). Even though the intensity of green fluorescence decreased as T cells proliferated, reliable signal:noise was maintained within the duration of co-culture (see below). To identify dead/dying cells, we loaded cells with a mixture of conjugated Annexin V and propidium iodide, dyes that fluoresce at red wavelengths, at the start of co-culture [14–16]. This dye mixture marked dead and dying cells with compromised membrane integrity. Accurate cell counting in the red channel required segmentation of the image under the following circumstances: (i) T cell cytotoxicity caused target cells to cluster; and, (ii) IncuCyte® lacked a third color channel to mark nuclei, a useful

independent observable for segmentation. Therefore, we used the total red pixel area to esti-mate the number of dead/dying cells. This dye imaging approach allowed us to computation-ally separate viable target cells, viable effector cells, and dead/dying target cells during co-culture, based on the unique image signature of each (Fig 1C).

To determine sensitivity and selectivity, we used a target-negative tumor cell line with pep-tide loading to control pMHC epitope number. As target cells, we chose adherent A375 mela-noma cells that are HPV-negative and HLA-A\*02-positive. The A375 cells were loaded with different peptide concentrations at the start of co-culture with T cells. A375 cells without loaded peptide served as a negative control, and allowed estimation of non-specific killing. By quantification of proliferation and death of target and T cells, the following parameters were computed to gain a better understanding of the process of T cell cytotoxicity: total target cell death, T cell proliferation, and T cell proliferation and kill rates/cell, as a function of peptide dose. We calibrated the instrument for fluorescence detection prior to each experiment. Where possible (e.g., input T cell and target cell numbers), we confirmed that the instrument recorded accurate counts. Finally, we spot checked images by eye to further confirm accurate automated reporting.

## Specificity and sensitivity of T cell cytotoxicity and proliferation

Specificity and sensitivity are among the most important attributes of engineered T cells, and their assessment can be confounded by several factors, including off-target activity of the engi-neered receptor, as well as alloreactivity of the receptor and/or expressed endogenous TCR repertoire. Initially, we compared 3 receptor constructs (2 TCRs and one CAR), all directed at HLA-A\*02 pMHC antigens (Table 1) [17, 18]. To determine parameters related to sensitivity and specificity, we first measured target cell confluency by phase contrast at different peptide concentrations. These raw data revealed slight differences in target cell growth with and with-out target peptide for the CAR and TCR constructs (Fig 2A, and S2 and S3 Figs). By focusing on T cells (labeled green), we could resolve their growth over time. Division by T cell count at t = 2 hours, after the T cells had settled to the well bottom, provided a correction for seeding differences in T cell number, showing proliferation rates on a per-cell basis. All receptors showed clear antigen-dependence. To estimate target-specific killing per T cell, we measured dead/dying target cells over time, and normalized to the total T cell count as well as to the con-dition of zero added peptide. The normalization yielded a series of kill vs. time curves, readily revealing differences among the three receptors.

Next, normalized data points corresponding to peptide dose-titration described above were fitted to a sigmoid curve. Peptide dose-responses were plotted for both T cell proliferation and killing at their maximum levels, 72 hrs. and 48 hrs., respectively (Fig 2B and 2C). The $EC_K50$ and $EC_P50s$ were similar for each construct and the TCR values were consistent with published results [17, 18]. Slightly higher proliferation levels ($E_Pmax$) were observed at lower E:T ratios. T cell proliferation measured by the IXM or flow cytometry displayed minor differences from the Incucyte®. On IXM, a slight decrease of T cell number at high (3:1) E:T ratio and 10 μM peptide concentration was observed (S4 Fig). These differences may be explained by: 1) The area imaged under 10x magnifying power on the IXM is significantly larger than IncuCyte®; 2) T cells tended to concentrate in the outer area in a well, especially when target cells were absent or cleared as a result of T cell cytotoxicity. Flow cytometry did not reveal higher prolif-eration at lower E:T ratios. However, it was challenging to compare T cell numbers among dif-ferent E:T ratios as the level of cell recovery was variable. On the other hand, flow cytometry confirmed that, at a fixed E:T ratio, T cells proliferated at a lower rate when they exhibited higher cytotoxicity (S5 Fig). In summary, proliferation sensitivity, $EC_P50$, was not substantially

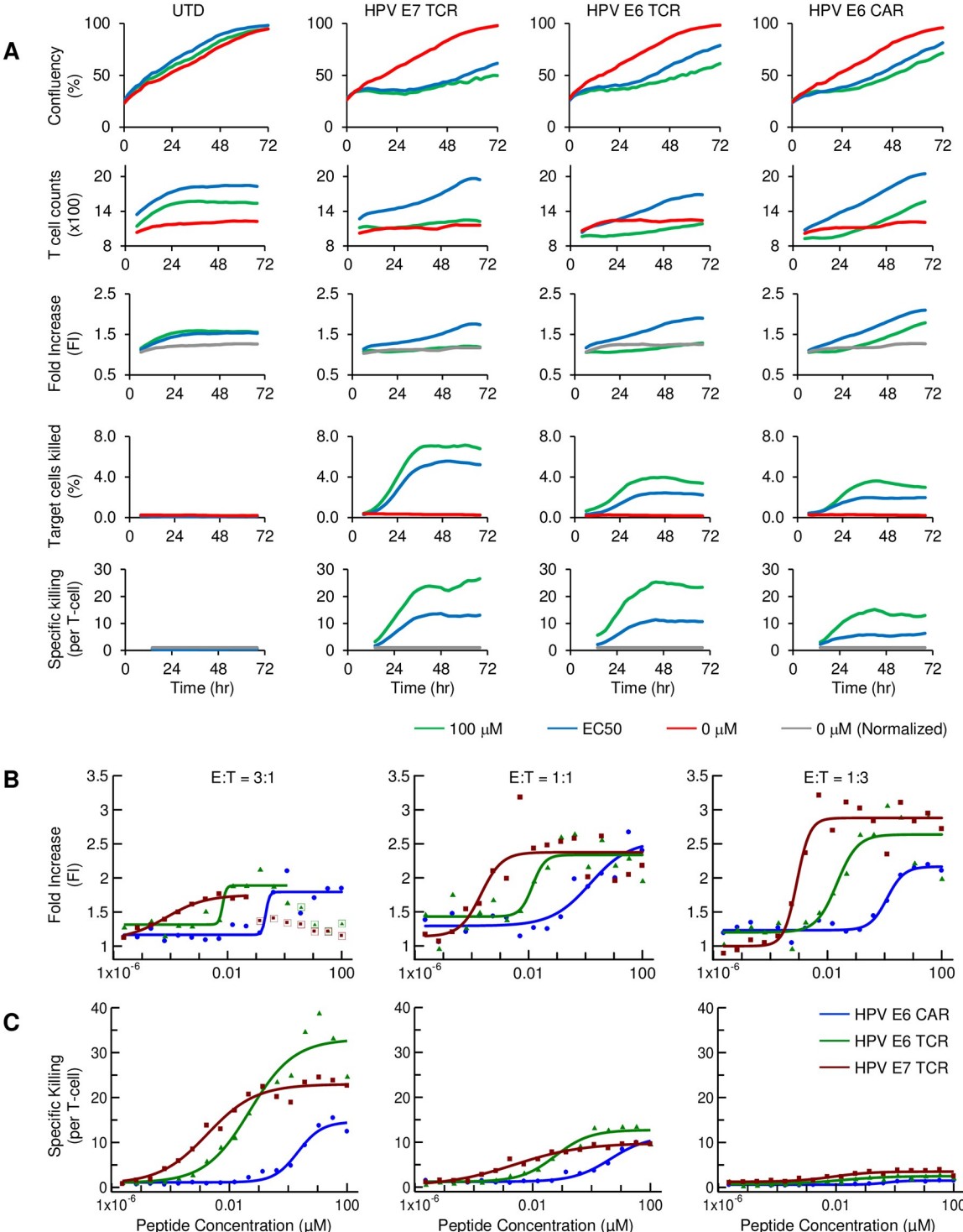

**Fig 2. Summary of T cell proliferation and specific killing.** (A) Time course for 3:1 E:T ratio of target cell confluency, T cell counts, relative T cell proliferation, percentage of target cells killed, and specific target cell killing per T cell, summarized for T cells that are either untransduced or transduced with one of the CAR/TCR constructs in target cells loaded with target peptide at 100 μM (green), 0 μM (red), or a concentration similar to their EC50 (blue) determined. Similar data for 1:1 and 1:3 in S2 and S3 Figs. (B) Dose-response of relative T cell proliferation; and, (C) specific killing. Measurements are made for both Fig 2A and 2B for all constructs at 3:1, 1:1, and 1:3 E:T ratios, at the time point when the peak value is reached: 72 hours for proliferation and 48 hours for killing.

**Table 2. Summary of parameters measured for constructs used in this study.**

| Construct | E:T ratio | Proliferation | | Specific killing | | Proliferation rate | | Kill rate | |
|---|---|---|---|---|---|---|---|---|---|
| | | EC50 (μM) | Emax (FI) | EC50 (μM) | Emax (T cell$^{-1}$) | EC50 (μM) | Emax (%·Day$^{-1}$) | EC50 (μM) | Emax (T cell$^{-1}$·hr$^{-1}$) |
| HPV E6 TCR | 3:1 | 0.007 | 1.79 | 0.049 | 32.99 | 0.0008 | 25.68 | 0.066 | 10.71 |
| | 1:1 | 0.014 | 2.54 | 0.059 | 12.73 | 0.0194 | 40.44 | 0.052 | 8.19 |
| | 1:3 | 0.022 | 2.64 | 0.016 | 2.51 | 0.0048 | 55.72 | 0.015 | 6.41 |
| HPV E6 CAR | 3:1 | 0.219 | 1.89 | 2.13 | 14.61 | 0.1093 | 29.51 | 4.747 | 7.54 |
| | 1:1 | 1.478 | 2.43 | 4.372 | 11.32 | 1.4222 | 53.69 | 3.356 | 5.94 |
| | 1:3 | 1.217 | 2.17 | 0.474 | 1.51 | 2.7469 | 58.69 | 0.143 | 2.53 |
| HPV E7 TCR | 3:1 | 0.00007 | 1.75 | 0.002 | 22.96 | 0.0001 | 3.72 | 0.0027 | 8.17 |
| | 1:1 | 0.0002 | 2.38 | 0.002 | 9.81 | 0.0002 | 40.92 | 0.0014 | 19.5 |
| | 1:3 | 0.001 | 2.88 | 0.008 | 3.52 | 0.0007 | 65.32 | 0.0028 | 7.76 |

affected by E:T ratio. Though maximum kill ($E_K$max) decreased with lower E:T ratio, the sensitivity of both TCRs and the CAR measured by specific killing were within a few fold of each other at all E:T ratios (Table 2).

## T cell cytotoxicity and proliferation rates

We used the proliferation and cytotoxicity data to compute proliferation and kill rates over time vs. peptide concentration (Fig 3; Methods). $EC_{P'}50$ and $EC_{K'}50$ were approximately the same for each construct. Moreover, E:T ratio did not substantially affect either parameter (Table 2).

Other features differed between the kill and proliferation rate curves. Proliferation rates lagged behind cytotoxicity and showed an optimal time and peptide concentration (Fig 4A).

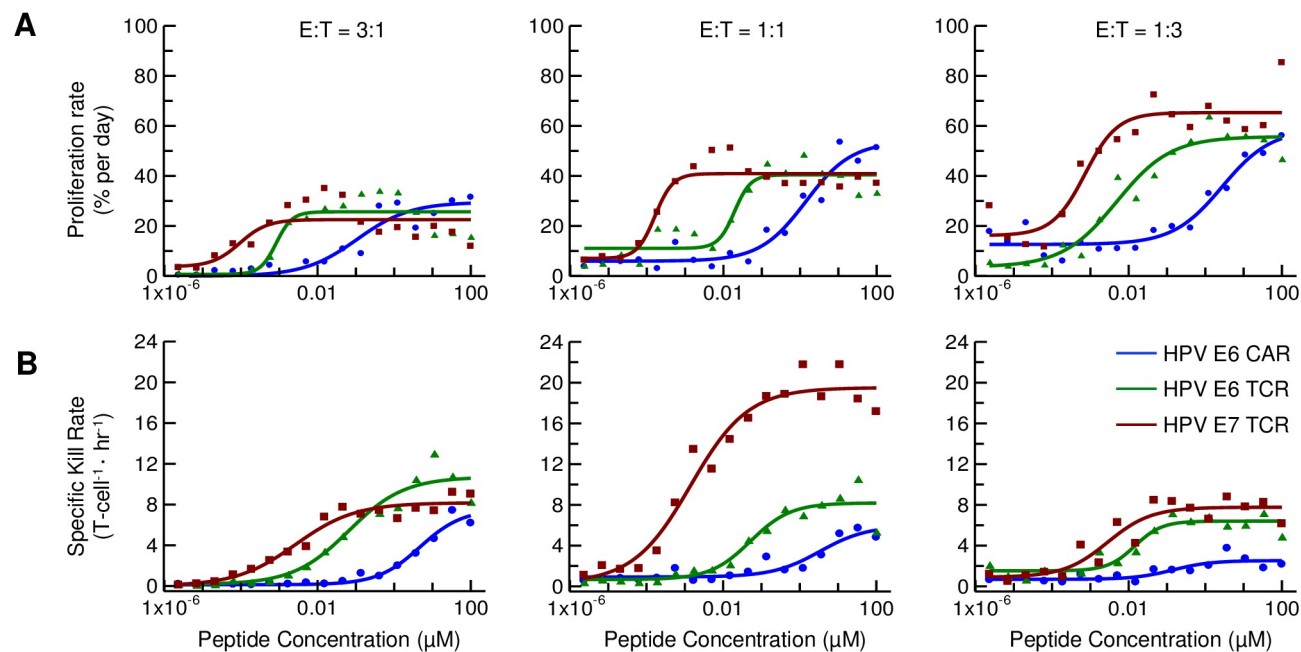

**Fig 3. Summary of sensitivity (dose-response) of T cell proliferation and kill rates.** Time point for dose-response chosen to be maximum rate for each peptide concentration between t = 30–72 hours; maximum rate is plotted vs. peptide dose. (A) Normalized proliferation rate of each construct. (B) Specific kill rate.

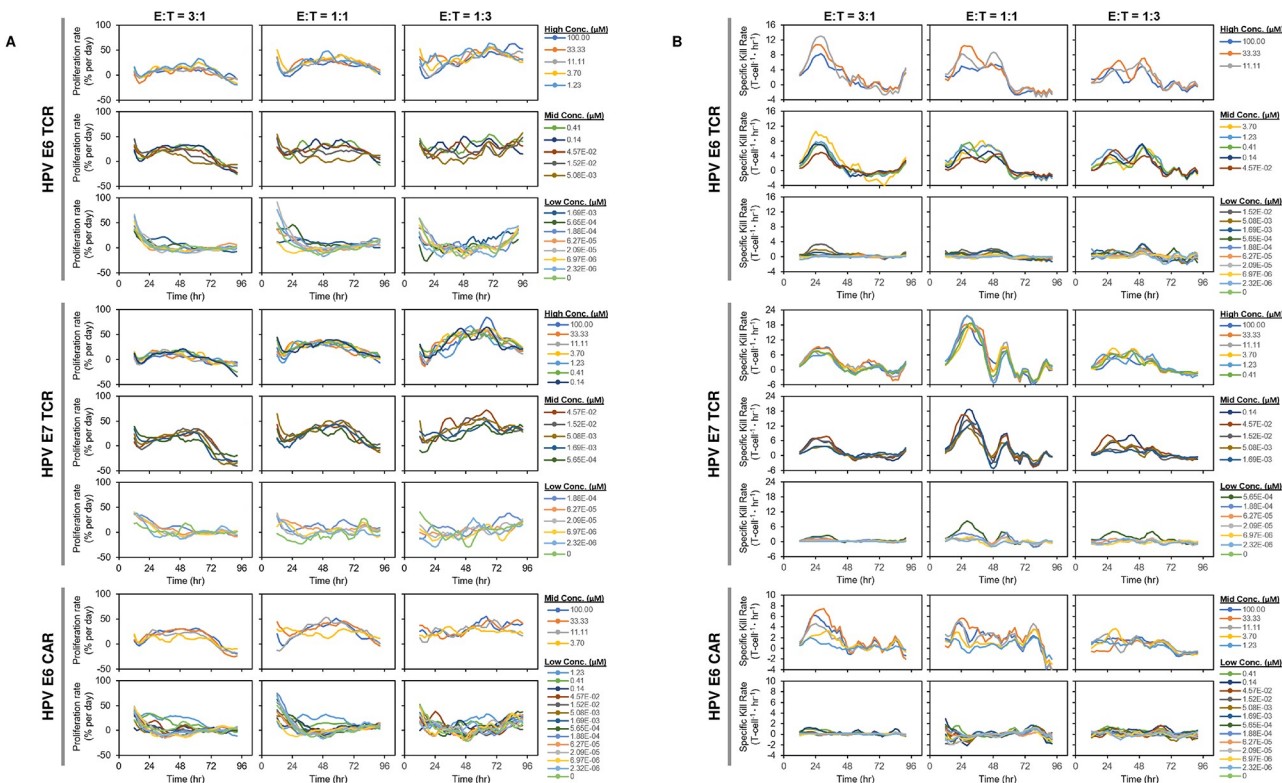

**Fig 4. Summary of kinetic profile of T cell proliferation and specific kill rates.** Time course plots were divided into three groups: 1) High: [peptide] >100x EC50; 2) Mid: 100x EC50 > [peptide] >1x EC50; 3) Low: [peptide] < 1x EC50. When proliferation time courses (A) are plotted, each group shows a different time dependent profile from the others. The highest proliferation is observed in the 2nd group. In the specific killing time course (B), all groups follow similar time course patterns. Specific killing peaks occur sooner than T cell proliferation peaks.

Interestingly, near the optimal peptide concentration for maximum proliferation, each proliferating T cell construct showed an initial dip in rate, followed by a steady increase before a final decline. In contrast, kill-rate vs. time plots revealed no initial decrease, but rather, increased steeply from the first measurable time point (8 hrs; Fig 4B). The maximum kill rate occurred at ~24 hours, for all constructs regardless of peptide concentration. There was a striking reproducible fine structure to some of the kill rate curves, suggesting periodicity in the kill rate over time. The pattern was clearest with the HPV E7 TCR and at lower E:T ratios, with a defined secondary peak at approximately 56 hours, followed by peaks at 72 and 88 hours, but falling over time. Though the time to peak of the kill rate (24 hour) was similar at all peptide concentrations sufficient to trigger a response, the magnitude of the rate was higher at higher peptide concentrations.

## Assay comparison: Example of the HPV E7 CAR

We note that typical assays used to assess CARs and TCRs do not distinguish effects caused by increased per-cell cytotoxicity from proliferation. The QuIK assay separates these variables. To demonstrate the utility of the QuIK assay in one simple case, we compared HPV E7 CAR against the equivalent TCR (HPV E7 TCR; Fig 5). Both receptors showed strong dose-dependent activation in a T2/Jurkat assay and cytotoxicity in conventional E:T-ratio assays (Fig 5A and 5B). In two assays—a killing assay using luciferase as a readout for viability and an IFNγ

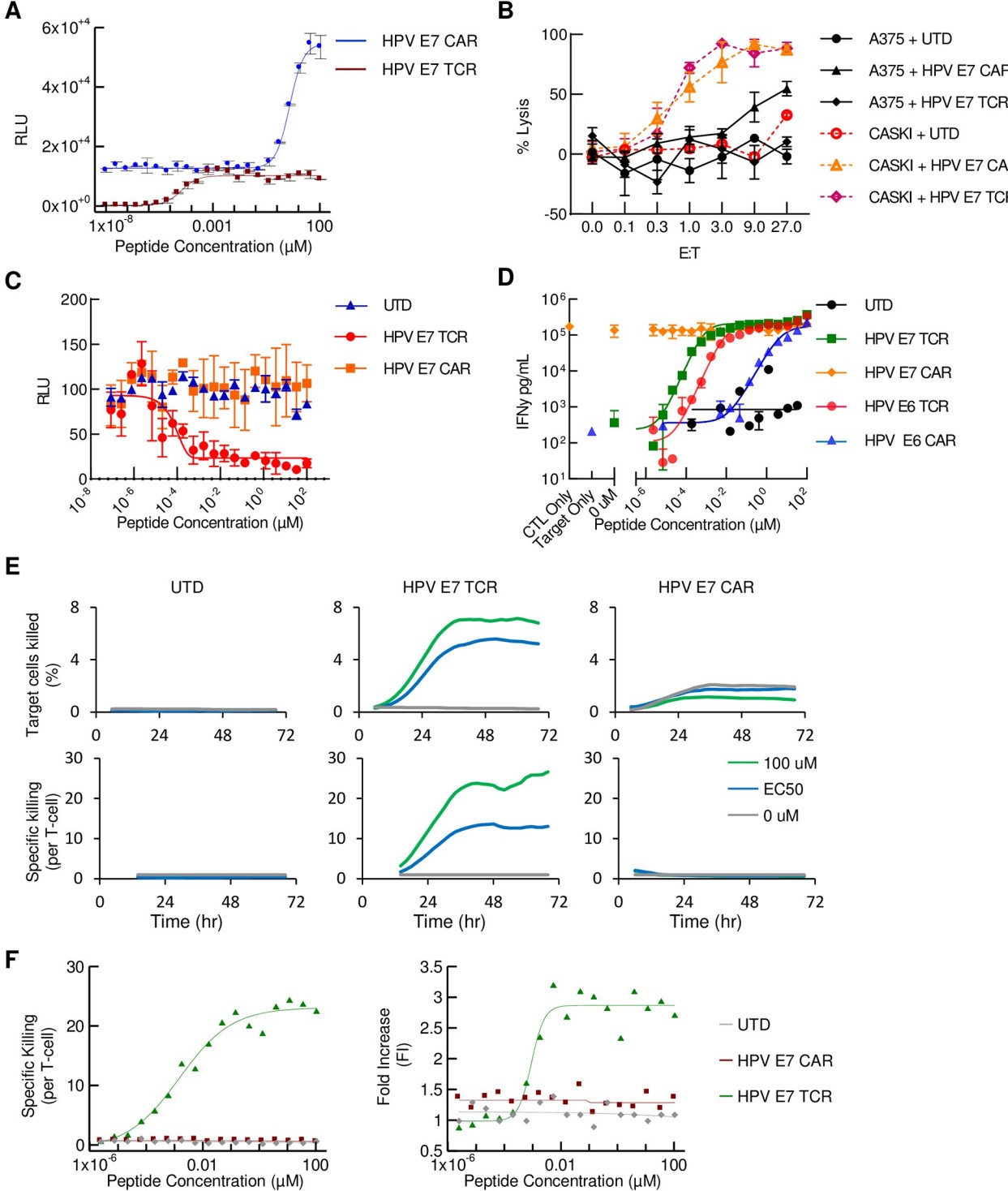

**Fig 5. Example of non-specific activity determined by QuIK assay.** (A) Jurkat/T2 assay. (B) E:T ratio assays with primary T cells and either A375 (HPV-negative) or CaSki (HPV-positive) as target cells; UTD, untransduced. (C) Peptide-loading, luciferase viability assay. (D) IFNγ as a readout for T cell activation. The signal was measured with AlphaLISA as described in Materials and Methods. T cells-only and T cells co-cultured with A375 without loaded peptide were used to determine target-independent activation. (E) Total target cell death and specific kill/cell as described in Results and Materials and Methods. (F) QuIK assay showing on-target cell killing and T cell proliferation of HPV E7 TCR but not HPV E7 CAR.

secretion assay—HPV E7 CAR did not demonstrate peptide-dose-dependent killing or activation (Fig 5C and 5D). However, in a single experiment, the QuIK assay readily revealed that HPV E7 CAR lacks both specific killing and proliferation (Fig 5E and 5F).

A limitation of the QuIK assay as currently configured is the inability to measure baseline or tonic signaling, an important parameter that may impact antigen-dependent function over time. Therefore, we used a conventional IFNγ measurement to accrue tonic-signaling data for the 4 receptors studied here (Fig 5D). The results showed that the HPV E7 CAR has very high tonic (target cell independent) activation, providing a mechanistic explanation for its different behavior.

## Discussion

T cells, the basis for current FDA-approved CAR-T cell therapies, are complex cellular machines that evolved to proliferate in response to specific antigens and differentiate into various effector subtypes, including cytotoxic T cells, specialized for antigen-dependent killing of target cells. Despite their clinical relevance, most *in vitro* assays of T cell function remain in a fairly primitive state, especially with regard to what are arguably the most relevant clinical features of T cell function, on-target cytotoxicity and proliferation vs. time.

The study of TCRs in particular has benefited enormously from a simple cellular assay that employs the Jurkat T cell line as effector and the T2 lymphoblast hybridoma line as stimulus [19]. The Jurkat/T2 NFAT-luciferase assay is robust, quantitative, rapid, and simple. It can be used to generate dose-response curves that reveal tonic signaling, general cellular off-target signaling, sensitivity of response (EC50), and maximal response (Emax). All of these parameters have relevance to the receptor under study; for instance, EC50 relates to the probability that the receptor will engage and be activated by a pMHC ligand on the T2 surface; and Emax relates in part to the level of functional receptor protein [20, 21].

We focused here on primary human T cell proliferation and cytotoxicity, fundamental consequences of T cell activation upon interaction with antigen. Even though T cell proliferation and cytotoxicity are believed to be closely related to each other, the detailed temporal relation between them is not fully understood, and rarely deconvoluted in kinetic assays. The QuIK assay enabled us to monitor these two important features of T cell activation independently and simultaneously in the same well over time. We were thus able to calculate T cell proliferation rate on an average-cell basis for both TCRs and CARs. We demonstrated the QuIK assay can also be performed on different imaging instruments, such as the IXM. T cell cytotoxicity observations from the IncuCyte® and IXM were consistent. T cell proliferation observed on these instruments followed a similar trend with small differences, most likely due to the difference in the size of their fields of view. Although we optimized the assay for A375 cells, we have used other cell lines successfully as well.

The data, so far limited to 2 CARs and TCRs, are consistent with the view that at short time intervals post stimulation, the rates of proliferation and cytotoxicity are inversely correlated at a given time in the heterogeneous T cell population. The basis for this observation in the QuIK assay remains to be explored; for instance, whether or not responses of two (or more) phenotypically distinct cell populations sum under these conditions to produce the net effects, as some have proposed for virus-specific CD8+ cells, and naïve and memory T cells in general [22, 23]. We noted in the data the possibility that killing is periodic, occurring in waves. However, we did not observe such periodicity in the proliferation data, perhaps a consequence of lower signal:noise in the proliferation measurements. Importantly, the sensitivities of the proliferation and cytotoxicity response revealed by the QuIK assay, including rates, were very similar for each receptor, consistent with the view that both killing and proliferation EC50s are

dictated primarily by the signaling competence of the receptor itself. Under the conditions of the QuIK assay, we saw no evidence for different activation thresholds for proliferation vs. cytotoxicity, as has been observed in other situations [24–26].

The assay revealed other potentially meaningful quantitative features, including the impact of E:T ratio on proliferation and cytotoxicity. E:T ratio had little effect either on the sensitivity of intrinsic killing and proliferation on a per-cell basis, or on the respective rates. However the magnitude of the response (Emax) was influenced by the E:T cell ratio on a per T cell basis (Table 2). Again, proliferation and cytotoxicity were inversely correlated: T cells proliferated more when the cytotoxicity was lower. We observed this phenomenon on the IncuCyte® where lower E:T ratios showed higher proliferation and lower cytotoxicity. Higher T cell proliferation at lower E:T ratios was less dramatic on the IXM (S4 Fig). However, all E:T ratios showed lower proliferation and higher cytotoxicity at 10 uM peptide compared to lower peptide concentrations. This observation was confirmed by flow cytometry to determine T cell proliferation (S5 Fig). These results raise questions about the interpretation of simpler assays that depend on high E:T ratios to achieve sufficient signal.

Assays like QuIK that separate conflated variables have proved invaluable to science. In the realm of drug development, a good example is medicinal chemistry's reliance on structure-activity relationships (SAR). SAR experiments with reliable assays have allowed chemists to interpret small quantitative differences among a family of molecules to optimize a variety of properties, including potency and selectivity. Similarly, as the CAR- and TCR-T fields mature, we expect that more detailed SAR approaches will prove critical to understand the structural basis for activity of T cells. It is important to bear in mind that the outputs of *in vitro* models of primary T cell function may depend on the conditions and donors used; however, the results presented here were generally consistent for other donors and other commonly used cell culture conditions; e.g., the presence of IL-15/IL-21. With the limited set of CARs and TCRs used here, it is difficult to draw extensive conclusions, but it seems likely, given the robust quality of QuIK assay data, that larger datasets will help differentiate TCRs and CAR designs, as well as provide a foundation for better prediction of functional properties. We view the QuIK assay as a step along the path to more predictive assays, perhaps of increased complexity; e.g., where T cell marker phenotypes are monitored over time, in addition to effector function. We believe that the vast majority of assays in current usage lack the capability to rank order receptors in a simple, consistent fashion, so much so that crude, xenogeneic *in vivo* models are often employed to select clinical candidates [27]. Thus, the QuIK assay may help provide insights into, and prioritization for, the next generation of TCR- and CAR-T therapies with regard to their characteristics, including therapeutic window. The assay, with its capacity to separate variables presumed to be important for clinical performance of engineered T cells, may also prove useful to qualify manufactured T cell products more effectively.

## Supporting information

**S1 Fig. Expression levels of TCR and CAR constructs, shown by flow cytometry.** (TIF)

**S2 Fig. Time course data for 1:1 E:T ratio on IncuCyte®.** Graphs of target cell confluency, T cell counts, relative T cell proliferation, percentage of target cells killed, and specific target cell killing per T cell are summarized here. Target cells are loaded with peptide at 100 uM (green), 0 uM (red), or a concentration similar to their EC50 (blue). (TIF)

**S3 Fig. Time course data for 1:3 E:T ratio on IncuCyte®.** Graphs of target cell confluency, T cell counts, relative T cell proliferation, percentage of target cells killed, and specific target cell killing per T cell are summarized here. Target cells are loaded with peptide at 100 uM (green), 0 uM (red), or a concentration similar to their EC50 (blue).
(TIF)

**S4 Fig. Time-course data for T cell proliferation and specific cytotoxicity on ImageXpress® Micro.** Graphs of relative T cell proliferation and specific target cell killing per T cell from three different E:T ratios are summarized. Target cells were loaded with peptide at 10 uM (green), 0 uM (gray), or a concentration similar to their EC50 (blue).
(TIF)

**S5 Fig. T cell proliferation determined by flow cytometry.** Method and results of determining T cell proliferation are summarized (see Materials and methods for further details). A. An example contour plot to illustrate how TCR or CAR-transduced T cells were separated from the untransduced T cells and target cells. B. Scatter plots of T cell flow cytometry at day 0 and day 3. C. Comparison of specific killing from IXM (red) and T cell proliferation (blue) determined by flow cytometry revealed an inverse relationship between cytotoxicity and T cell proliferation.
(TIF)

## Acknowledgments

We thank Drs. Mark L. Sandberg, Jing-Ping Hsin, and Aaron D. Martin for T cell culture, Michele E. McElvain for Jurkat and T2 cell validation, Dr. Chuck Li for detailed imaging analysis, and Drs. Will Go and Michelle Kreke for discussion and comments on the manuscript.

## Author Contributions

**Conceptualization:** Grace L. Huang, Daniel P. Nampe, Alexander Kamb, Han Xu.

**Formal analysis:** Grace L. Huang, Daniel P. Nampe.

**Investigation:** Grace L. Huang, Daniel P. Nampe, Jason Yi, Grant B. Gabrelow, Kathleen R. Negri.

**Methodology:** Grace L. Huang, Daniel P. Nampe, Jason Yi, Alexander Kamb, Han Xu.

**Supervision:** Alexander Kamb, Han Xu.

**Validation:** Grace L. Huang, Daniel P. Nampe, Han Xu.

**Visualization:** Grace L. Huang, Daniel P. Nampe, Han Xu.

**Writing – original draft:** Grace L. Huang, Daniel P. Nampe, Alexander Kamb, Han Xu.

**Writing – review & editing:** Grace L. Huang, Daniel P. Nampe, Alexander Kamb, Han Xu.

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
