## [Decision Letter · Decision Letter 0]

17 Aug 2020

PONE-D-20-12964

A multivariate, quantitative assay that disentangles key kinetic parameters of primary human T cell function in vitro

PLOS ONE

Dear Dr. Xu,

Thank you for submitting your manuscript to PLOS ONE. After careful consideration, we feel that it has merit but does not fully meet PLOS ONE’s publication criteria as it currently stands. Therefore, we invite you to submit a revised version of the manuscript that addresses the points raised during the review process.

Please make necessary changes suggested by reviewers.

We look forward to receiving your revised manuscript.

Kind regards,

Nupur Gangopadhyay, B.V.Sc, M.V.Sc.,Ph.D.

Academic Editor

PLOS ONE

Journal Requirements:

"GH, DN, KN, GG, AK, and HX are employees of A2 Biotherapeutics

JY is an employee of ImmPACT-Bio."

We note that one or more of the authors are employed by a commercial company: A2 Biotherapeutics, ImmPACT-Bio.

2.1. Please provide an amended Funding Statement declaring this commercial affiliation, as well as a statement regarding the Role of Funders in your study. If the funding organization did not play a role in the study design, data collection and analysis, decision to publish, or preparation of the manuscript and only provided financial support in the form of authors' salaries and/or research materials, please review your statements relating to the author contributions, and ensure you have specifically and accurately indicated the role(s) that these authors had in your study. You can update author roles in the Author Contributions section of the online submission form.

2.2. Please also provide an updated Competing Interests Statement declaring this commercial affiliation along with any other relevant declarations relating to employment, consultancy, patents, products in development, or marketed products, etc.  

Reviewers' comments:

Reviewer's Responses to Questions

**Comments to the Author**

1. Is the manuscript technically sound, and do the data support the conclusions?

Reviewer #1: Yes

Reviewer #2: Yes

2. Has the statistical analysis been performed appropriately and rigorously? 

Reviewer #1: Yes

Reviewer #2: Yes

3. Have the authors made all data underlying the findings in their manuscript fully available?

Reviewer #1: Yes

Reviewer #2: Yes

4. Is the manuscript presented in an intelligible fashion and written in standard English?

Reviewer #1: Yes

Reviewer #2: Yes

5. Review Comments to the Author

Reviewer #1: Huang et al report a multivariate, quantitative assay to address kinetic measures of human T cells functional tests. Imagining based T cell functional test offers several advantage over classical methods. This method records time and antigen dependent T cell proliferation and cytotoxicity at several effector to target ratio. Informed consent and institutional guidelines, and HIPPA compliance were followed to conduct this study.

T2, Jurkat, and HPV-negative HLA-A2 positive target cell lines were used. Figure 1- Schema of T cell transduction defines experimental design.

Table 1: Only HLA-A2 restricted constructs are used and antigen peptides are shown. Table 2: A detail summary of measured parameters for constructs are shown.

A limitataion to Qulk assay to its inability for baseline and tonic signaling is shown. HPVE7 CAR target cell independent killing is highlighted. Supporting information is well described and included FACS data for gene expression, Confluency, T cell#, Target cell killing, and specific killing.

Overall this study is well designed and nicely written with enough details for replication purpose.

Reviewer #2: The manuscript titled “A multivariate, quantitative assay that disentangles key kinetic parameters of primary human T cell function in vitro” by Huang et al. addresses a key technical gap in the T cell field: how to separate the key functional parameters of time- and antigen- dependent T cell proliferation and cytotoxicity. This methodology will not only be useful for high content analysis of T cell function for adoptive therapies, but also for basic understanding of effector T cell biology using high-throughput assays. The authors establish a Quantitative Imaging-based Killing (QuIK) assay, where antigen-specific T cells kill rate can be distinguished from their proliferation rate. This assay revealed some important differences in cytotoxic capacity of cells bearing different antigen receptors. In my opinion, the manuscript is an important advance in the field of CAR- and TCR- T cell therapy.

My only question is regarding one of the experiments (Figure 2B,C) where authors show that the T cell proliferation and killing are reciprocally linked, and show different values at different E:T ratios. For example, proliferation is lower at high E:T (3:1) and higher at low E:T (1:3) whereas Killing follows an opposite behavior. Are these behaviors assay dependent? Or this is also seen outside of QuIK assay. My suggestion is to measure these parameters independently to assess that, and discuss a possible explanation underlying this behavior in the Discussion section of the manuscript.

In addition, a minor suggestion is to change the color scheme of the traces in Figure 2B, C and Figure 3 to allow them to be more distinguishable.

6. PLOS authors have the option to publish the peer review history of their article (what does this mean?). If published, this will include your full peer review and any attached files.

Reviewer #1: **Yes: **Raghvendra M. Srivastava, PhD

Reviewer #2: No

---

## [Author Response · Author response to Decision Letter 0]

12 Oct 2020

PONE-D-20-12964

A multivariate, quantitative assay that disentangles key kinetic parameters of primary human T cell function in vitro

PLOS ONE

 We have formatted the manuscript according to the guidelines.

"GH, DN, KN, GG, AK, and HX are employees of A2 Biotherapeutics

JY is an employee of ImmPACT-Bio."

We note that one or more of the authors are employed by a commercial company: A2 Biotherapeutics, ImmPACT-Bio.

2.1. Please provide an amended Funding Statement declaring this commercial affiliation, as well as a statement regarding the Role of Funders in your study. If the funding organization did not play a role in the study design, data collection and analysis, decision to publish, or preparation of the manuscript and only provided financial support in the form of authors' salaries and/or research materials, please review your statements relating to the author contributions, and ensure you have specifically and accurately indicated the role(s) that these authors had in your study. You can update author roles in the Author Contributions section of the online submission form.

As requested, we have added the following statement: The funder provided support in the form of salaries for authors GLH, DPN, GBG, KRN, AK, and HX, but did not have any additional role in the study design, data collection and analysis, decision to publish, or preparation of the manuscript. The specific roles of these authors are articulated in the ‘author contributions’ section.

2.2. Please also provide an updated Competing Interests Statement declaring this commercial affiliation along with any other relevant declarations relating to employment, consultancy, patents, products in development, or marketed products, etc. 

As requested, we have included the statement: "This does not alter our adherence to PLOS ONE policies on sharing data and materials.” 

We have provided the amended Funding and Competing Interests statements in the cover letter per request and thank the editor in advance for altering the online submission.

Reviewers' comments:

Reviewer's Responses to Questions

Comments to the Author

1. Is the manuscript technically sound, and do the data support the conclusions?

Reviewer #1: Yes

Reviewer #2: Yes

2. Has the statistical analysis been performed appropriately and rigorously?

Reviewer #1: Yes

Reviewer #2: Yes

3. Have the authors made all data underlying the findings in their manuscript fully available?

Reviewer #1: Yes

Reviewer #2: Yes

4. Is the manuscript presented in an intelligible fashion and written in standard English?

Reviewer #1: Yes

Reviewer #2: Yes

5. Review Comments to the Author

Reviewer #1: Huang et al report a multivariate, quantitative assay to address kinetic measures of human T cells functional tests. Imagining based T cell functional test offers several advantage over classical methods. This method records time and antigen dependent T cell proliferation and cytotoxicity at several effector to target ratio. Informed consent and institutional guidelines, and HIPPA compliance were followed to conduct this study.

T2, Jurkat, and HPV-negative HLA-A2 positive target cell lines were used. Figure 1- Schema of T cell transduction defines experimental design.

Table 1: Only HLA-A2 restricted constructs are used and antigen peptides are shown. Table 2: A detail summary of measured parameters for constructs are shown.

A limitataion to Qulk assay to its inability for baseline and tonic signaling is shown. HPVE7 CAR target cell independent killing is highlighted. Supporting information is well described and included FACS data for gene expression, Confluency, T cell#, Target cell killing, and specific killing.

Overall this study is well designed and nicely written with enough details for replication purpose.

Reviewer #2: The manuscript titled “A multivariate, quantitative assay that disentangles key kinetic parameters of primary human T cell function in vitro” by Huang et al. addresses a key technical gap in the T cell field: how to separate the key functional parameters of time- and antigen- dependent T cell proliferation and cytotoxicity. This methodology will not only be useful for high content analysis of T cell function for adoptive therapies, but also for basic understanding of effector T cell biology using high-throughput assays. The authors establish a Quantitative Imaging-based Killing (QuIK) assay, where antigen-specific T cells kill rate can be distinguished from their proliferation rate. This assay revealed some important differences in cytotoxic capacity of cells bearing different antigen receptors. In my opinion, the manuscript is an important advance in the field of CAR- and TCR- T cell therapy.

My only question is regarding one of the experiments (Figure 2B,C) where authors show that the T cell proliferation and killing are reciprocally linked, and show different values at different E:T ratios. For example, proliferation is lower at high E:T (3:1) and higher at low E:T (1:3) whereas Killing follows an opposite behavior. Are these behaviors assay dependent? Or this is also seen outside of QuIK assay. My suggestion is to measure these parameters independently to assess that, and discuss a possible explanation underlying this behavior in the Discussion section of the manuscript.

We are very grateful to the reviewer for pointing out this inconsistency, and we have addressed it in the manuscript. We used an orthogonal method, flow cytometry, to measure proliferation (S5 Fig.). Even though it was challenging to compare among different E:T ratio groups due to T cell recovery, we demonstrated the inverse correlation between cytotoxicity and proliferation within each E:T ratio. In the meantime, we acquired a more sophisticated (and expensive) imaging instrument called the ImageXpress Micro (IXM). When we compared results between the Incucyte and IXM, we noticed that at high E:T ratios, the Incucyte systematically underestimates T cell counts, because of the tendency for T cells at high concentration to aggregate in the periphery of wells. The Incucyte can only image the well center; the IXM images the whole well. Even with this difference, we observed a similar trend to Incucyte results with slight differences: 1) the T cell proliferation was similar across different E:T ratio groups (all are very modest), with the exception that, for the 3:1 group at the highest peptide concentration, total T cell numbers decreased over time. This result does not change the fundamental conclusions. However, it does suggest that a more expensive device has advantages.

We have altered the manuscript text to reflect these conclusions; specifically, we have:

1. added a section of flow cytometry to Methods (line 111-125);

2. added a sentence in Methods (QuIK Assay section) to reflect that the experiments can be done on both the IncuCyte and IXM (line 202);

3. added the following lines in Results to include the new data:

a. Line 296-298

b. Line 373-383

4. changed the following in Discussion:

a. Line 474-477

b. Line 499-505

5. added figures S4 and S5 to show data from the IXM and flow cytometry experiments.

In addition, a minor suggestion is to change the color scheme of the traces in Figure 2B, C and Figure 3 to allow them to be more distinguishable.

We have improved the clarity by increasing the line thickness.

6. PLOS authors have the option to publish the peer review history of their article. If published, this will include your full peer review and any attached files.

Do you want your identity to be public for this peer review? For information about this choice, including consent withdrawal, please see our Privacy Policy.

Reviewer #1: Yes: Raghvendra M. Srivastava, PhD

Reviewer #2: No

---

## [Editor Report · Decision Letter 1]

15 Oct 2020

A multivariate, quantitative assay that disentangles key kinetic parameters of primary human T cell function in vitro

PONE-D-20-12964R1

Dear Dr. Xu,

We’re pleased to inform you that your manuscript has been judged scientifically suitable for publication and will be formally accepted for publication once it meets all outstanding technical requirements.

Kind regards,

Nupur Gangopadhyay, B.V.Sc, M.V.Sc.,Ph.D.

Academic Editor

PLOS ONE
---

## [Editor Report · Acceptance letter]

28 Oct 2020

PONE-D-20-12964R1 

A multivariate, quantitative assay that disentangles key kinetic parameters of primary human T cell function *in vitro*

Dear Dr. Xu:

I'm pleased to inform you that your manuscript has been deemed suitable for publication in PLOS ONE. Congratulations! Your manuscript is now with our production department. 

Kind regards, 

on behalf of

Dr Nupur Gangopadhyay 

Academic Editor

PLOS ONE